# Extracellular Vesicles in Hepatocellular Carcinoma: Progress and Challenges in the Translation from the Laboratory to Clinic

**DOI:** 10.3390/medicina59091599

**Published:** 2023-09-05

**Authors:** Rong Yan, Haiming Chen, Florin M. Selaru

**Affiliations:** 1Department of Surgical Oncology, the First Affiliated Hospital, Xi’an Jiaotong University College of Medicine, Xi’an 710061, China; 2Division of Gastroenterology and Hepatology, Johns Hopkins University School of Medicine, Baltimore, MD 21224, USA; hchen10@jhmi.edu; 3Department of Oncology, Sidney Kimmel Cancer Center, Johns Hopkins University, Baltimore, MD 21224, USA; 4The Institute for Nanobiotechnology, The Johns Hopkins University, Baltimore, MD 21231, USA

**Keywords:** hepatocellular carcinoma, extracellular vesicle, diagnostic markers, therapeutic carrier

## Abstract

Extracellular vesicles (EVs) play critical roles in intercellular communication by transporting bioactive cargo to recipient cells. EVs have been implicated in a range of physiological and pathological processes, including tumor progression, metastasis, immune modulation, and drug resistance. The objective of this review is to present a thorough overview of recent studies focusing on EVs in hepatocellular carcinoma (HCC), with an emphasis on their potential utility as diagnostic biomarkers as well as therapeutic agents. Initially, we explore the utility of EVs as diagnostic biomarkers for HCC, followed by a discussion of their potential as carriers of therapeutic payloads. Additionally, we delve into the emerging field of therapeutic EVs for modulating tumor immune responses. Through this review, our ultimate aim is to provide a comprehensive understanding of the opportunities and challenges in the clinical translation of EV research in the domain of HCC.

## 1. Background

Extracellular vesicles (EVs) are a heterogenous group of membrane-bound particles that transport bioactive molecules, such as metabolites, lipids, proteins, and nucleic acids, to communicate with recipient cells [1]. The membranes of EVs protect their internal cargo and express specific surface ligands that enable efficient binding to recipient cells [2]. Through the transfer of their cargoes to recipient cells, EVs participate in various processes, including tumor initiation and formation [3], apoptosis [4], angiogenesis [5], metastasis [6], immune escape [7], and drug resistance [8].

EVs are broadly categorized as exosomes, microvesicles, and apoptotic bodies, based on their size and biogenesis [9]. Exosomes have a diameter ranging from 30 to 100 nm, and are believed to originate from intracellular multivesicular bodies before being secreted though the plasma membrane [10]. Microvesicles, also known as ectosomes, are sized from 100 to 1000 nm, and are derived from the plasma membrane through budding [11]. Apoptotic bodies have diameters ranging from 1 to 4 µm, and are released during late-stage programmed cell death [12]. Notably, however, the current methods for isolating and characterizing EVs pose challenges in terms of distinguishing among these three types [13]. Therefore, in this review, we will generically utilize the EV terminology to describe all types of vesicles.

Hepatocellular carcinoma (HCC) is the most common type of liver cancer, accounting for 90% of all cases [14]. The early detection of HCC is difficult because patients often have few or no symptoms [15]. The current diagnostic methodologies for HCC include serum biomarker alpha-fetoprotein (AFP), ultrasonography, and other forms of imaging. These diagnostic tests, however, have low sensitivity for early detection [16]. Moreover, the presence of chronic liver disease (CLD) can further affect the accuracy of diagnosis [16]. To address these limitations, researchers have investigated alternative liquid biopsy biomarkers for high-risk populations [17]. EVs are emerging as promising candidates since they are secreted in greater quantities by cancer cells compared to normal cells, contain abundant genetic materials that can be measured, and can be targeted with antibodies [18]. In addition to blood, EVs from different body fluids, such as urine and ascites, were used for tumor diagnosis, which may be important for non-invasive diagnostic and prognostic approaches.

EVs have a lipid bilayer structure that can effectively package hydrophobic and hydrophilic drugs. Another aspect relevant to therapeutics is the possibility of targeting EVs and their cargo. The surfaces of EVs express peptides, either from the cell of origin or engineered, that can specifically bind to receptors on target cells. As the liver is the most important organ involved in human metabolism, exogenous EVs tend to be spontaneously enriched in the liver. This enrichment pattern and the rapid clearance from the circulation by liver macrophages [19] make EVs appealing as drug delivery carriers to target liver diseases such as HCC [20]. On the other hand, HCC-derived EVs have well-established roles in facilitating tumor progression, metastasis, and unfavorable prognosis. They can also serve as molecular conduits to systemically disseminate oncogenic signals that drive key disease hallmarks like proliferation, invasion, metastasis, and therapeutic resistance.

## 2. The Role of EVs in the Diagnosis of HCC

Extracellular vesicles have emerged as promising biomarkers for cancer diagnosis. With some EV markers, such as ExoDxLung (ALK) [21] and ExoDxProstate IntelliScore, which have already been translated into successful clinical applications [22], there is growing interest in utilizing EVs as liquid biopsy tools. As sequencing technologies continue to advance, EV nucleic acid testing is expected to become one of the liquid biopsy technologies utilized for early diagnosis. It has the advantages of being minimally invasive, allowing for repeated sampling, and eliminating risks like bleeding, infection, and needle track seeding of cancer cells.

Several studies have demonstrated the potential of EV-carried miRNAs, specifically miR-21, as diagnostic biomarkers for HCC. The sensitivity of detecting miR-21 in EVs was found to be much higher than in serum alone [23,24,25]. For instance, Tomimaru et al. showed that while the total circulating expression of miR-21 was elevated in HCC patients, its abundance in EVs was much higher [25], indicating that EVs carrying miR-21 could be more sensitive diagnostic markers. Similarly, Jun et al. found that the miR-21 levels in EVs were significantly higher in the HCC patients than in those with chronic hepatitis B (2.12-fold) or the healthy individuals (5.57-fold) [26]. Another study found that miR-21 and miR-10b were significantly increased in EVs from HCC patients compared with healthy individuals or those with chronic hepatitis B, suggesting their potential as diagnostic biomarkers for early HCC [27]. Additionally, Pu et al. reported significant differences in the expression levels of EV miR-21-5p and miR-144-3p between HCC patients and those with chronic hepatitis B, with higher sensitivity and specificity than AFP in HCC diagnosis [28]. Notably, the EVs-miR-HCC Score system, which includes serum AFP, plasma EV miR-21-5p, and EV miR-92a-3p, has been shown to outperform AFP alone, with an area under the curve (AUC) of 0.85 compared to 0.72 for AFP-based HCC diagnosis, according to Sorop et al. [29].

MiR-122, an abundant EV cargo, has also shown promise as a diagnostic biomarker for HCC. Wang et al. found that the combination of serum EV miR-122, EV miR-148a, and AFP could distinguish early-stage HCC from cirrhosis with high accuracy, increasing the AUC to 0.931 (95% CI: 0.857–0.973) [30]. Similarly, an EV miRNA panel including miR-122, miR-148a, and miR-1246, was found to be significantly higher in HCC patients compared to cirrhotic and normal controls. When used together with AFP, the AUC increased to 0.931 regarding differentiating early HCC from cirrhosis, which was significantly better than using AFP alone (AUC = 0.712). Importantly, miR-122 has also shown high accuracy in distinguishing HCC from normal controls (AUC = 0.990) [31]. These results suggest that EV-associated miR-122 could be a valuable diagnostic biomarker for HCC, particularly in combination with other EV markers or the serum biomarker AFP.

In recent years, several other miRNAs carried by EVs have been identified as potential diagnostic biomarkers for HCC. For example, serum EV miR-4661-5p has shown good diagnostic performance at all stages of HCC, with an AUC = 0.917, even at early stages (AUC = 0.923), and has been found to be more accurate than other candidate serum miRNAs and serum AFP [32]. In addition, a panel of EV miRNAs consisting of miR-4661-5p and miR-4746-5p has been considered an accurate biomarker for the early diagnosis of HCC, with an AUC of 0.947. Sun et al. [33] found that serum EVs miR-331-3p were significantly up-regulated in patients with early-stage HCC and significantly decreased after surgical resection, while the serum EV miR-23b-3p was significantly increased in HCC patients after surgical resection. Similarly, Shi et al. [34] tested EV-carrying miRs from 126 HCC patients and 21 healthy controls, and found that the EV miR-638 levels were significantly lower in the HCC group, and the degree of reduction was negatively correlated with the liver cancer size and TNM stage.

Other miRNAs carried by EVs, such as miR-92b [35], miR-10b [36], miR-125b [37], miR-103 [38], miR-146a [39], miR-150-3p [40], miR-1307-5p [41], miR-221 [42], miR-665 [43], and miR-210 [44], have also been suggested as potential early diagnostic biomarkers. However, the reliability of these candidates needs to be further validated, as the studies supporting these claims have been conducted on small sample sizes. Therefore, further studies with larger sample sizes are required in order to confirm the diagnostic potential of these EV-associated miRNAs.

Recent studies have highlighted the potential of long non-coding RNAs (lncRNAs) isolated from serum EVs as non-invasive biomarkers for the diagnosis of HCC. These markers have advantages over AFP in distinguishing HCC patients from non-HCC patients. For example, Matboli et al. [45] reported that the EV lncRNA-P11-583F2.2 showed higher sensitivity and specificity (96.7%, 91.7%) than AFP (90%, 85%) in diagnosing HCC, and lncRNA JPX had a sensitivity of 91.5% and specificity of 72.7%. In another study, EV lncRNA-RP11-583F2.2 and lncRNA JPX were identified as upregulated in HCC and showed potential as diagnostic biomarkers for HCC with AUC values of 0.965 and 0.865, respectively [46]. Xu et al. [47] compared the expression levels of serum EV ENSG00000258332.1 and LINC00635 in HCC patients, hepatitis B patients, and healthy controls. They found that both markers were significantly higher in the HCC group and significantly lower after surgery. The combination of ENSG00000258332.1, LINC00635, and AFP had an AUC of 0.894, with a sensitivity of 83.6% and a specificity of 87.7%. Additionally, five lncRNAs associated with prognosis, including lncRNA-ATB [48], lncRNA CTD-2116N20.1 [49], lncRNA RP11538D16.2 [49], lncRNA CRNDE [50], and lncRNA ASMTL-AS1 [51], were found to be significantly upregulated in the serum EVs of HCC patients and associated with prognosis. For example, LncRNA CRNDE was overexpressed in HCC patients with large, poorly differentiated tumors and advanced TNM staging, and was an independent predictor of poor prognosis. Similarly, LncRNA ASMTL-AS1 expression was associated with HCC stage, metastasis, and prognosis. Elevated lncRNA-ATB expression suggested shorter overall survival (OS) and progression-free survival (PFS), and may also serve as an independent predictor of tumor progression or death. LncRNA CTD-2116N20.1 and lncRNA RP11538D16.2 were also overexpressed in HCC patients with shorter OS and PFS, and both contribute to poor prognoses of HCC patients by regulating protein expression levels in EVs.

Changes in circular RNA (circRNA) [52] and transfer RNA-derived small RNA (tsRNA) [53] expression have also been reported as potential diagnostic markers for HCC. Sun et al. [54] found that the expression levels of hsa_circ_0004001, hsa_circ_0004123, and hsa_circ_0075792 in the plasma EVs of HCC patients had higher diagnostic sensitivity and specificity and were positively correlated with TNM stage and tumor size. Similarly, Zhu et al. [53] compared the expression of tsRNAs in plasma EVs and found that 46 tsRNAs (35 up-regulated and 11 down-regulated) were differentially expressed in HCC patients and healthy donors. The expression levels of tRNA-ValTAC-3, tRNA-GlyTCC-5, tRNA-ValAAC-5, and tRNA-GluCTC5 were significantly higher in the plasma EVs of HCC patients compared with healthy donors, suggesting that plasma EV tsRNAs could serve as new diagnostic biomarkers. A recent study [55] used high-throughput sequencing technology to analyze the expression levels of PIWI-interacting RNAs (piRNAs) in serum EVs from 15 HCC patients and 15 controls. The findings from this analysis showed significant differential expression of piRNAs in serum EVs among HCC patients, indicating five possible biomarkers for the diagnosis of HCC. All five serum EV piRNAs demonstrated high AUROC values (piR-1029: 0.961; piR-15254: 0.868; novel-piR-35395: 0.898; novel-piR-32132: 0.926; novel-piR-43597: 0.935). Combining these five piRNAs (5-piRNA features) increased the AUROC value even further to 0.986.

Proteins have also been examined as another type of cargo in EVs in HCC patients. Certain proteins have been found to be elevated in EVs from patients with HCC, suggesting their potential as biomarkers for HCC diagnosis. For example, Arbelaiz A et al. [56] found elevated levels of RasGAP SH3 domain-binding protein (G3BP) and polymeric immunoglobulin receptor (PIGR) in the EVs of HCC patients, which had higher predictive efficacy for HCC than AFP. Similarly, Sun et al. [18] evaluated surface proteins of EVs and identified four potential HCC-associated protein markers, i.e., EpCAM, CD147, GPC3, and ASGPR1. These markers were able to differentiate early-stage HCC from cirrhosis, and were validated in a phase II case–control study involving 45 treatment-naïve patients with early-stage HCC, 23 patients with other cancers, and 61 patients with cirrhosis. This panel of markers showed an AUROC of 0.95 (95% CI, 0.90–0.99) with a sensitivity of 91% and a specificity of 90%. In addition, Fu et al. [57] demonstrated the presence of SMAD3-rich EVs in the peripheral blood of HCC patients, which exhibited a positive correlation with disease stage, tumor adhesion, pathological severity, and prognosis. A simultaneous combination test of both SMAD3-EVs and AFP resulted in a significant improvement in the diagnostic accuracy of human HCC, with an AUROC = 0.9. Furthermore, PKM2, a protein released from HCC cells, was found to promote HCC growth [58]. Plasma EV PKM2 levels were significantly upregulated in 34 HCC patients compared to healthy controls, suggesting its potential as a diagnostic marker for early HCC.

In conclusion, the studies reviewed above indicate the potential of EVs as a non-invasive diagnostic tool for HCC. EVs derived from HCC cells contain various cargoes, including miRNAs, circRNAs, tsRNAs, and proteins, which have been shown to have differential expression levels in HCC patients compared to healthy individuals. These promising findings suggest that EVs could serve as a tool for early HCC detection. A summary of EV-derived molecules used in the non-invasive diagnosis of HCC can be found in Table 1. Further research is needed to support the data and explore the full potential of EVs in HCC diagnosis and treatment.

## 3. EVs, the Next Generation of Carriers for Drug Delivery

Liposomal and polymeric nanoparticles (NPs) are being investigated intensely for their promise in delivering various types of drug molecules, for example, anticancer drugs, antifungal drugs, and analgesics. However, they have limitations, such as their inability to withstand changes in shear pressure, temperature, pH, or diluent concentration, as well as their relative inability to precisely deliver drugs to specific cell types in the body [59]. EVs offer a promising alternative to liposomal and polymeric drug delivery systems due to their long circulating half-life, biocompatibility, lack of intrinsic toxicity, and ability to target tissues [60]. Some clinical applications of EVs with drug-loaded capabilities have shown therapeutic efficacy in pancreatic cancer, such as mesenchymal stem cell (MSC)-derived EVs loaded with K-RAS G12D-specific siRNAs, which have extended the half-life and targeting of nucleic acid drugs and are now in clinical trials [NCT03608631] [61]. By engineering EVs to express specific surface proteins, glycans, peptides, or charges, researchers also aim to optimize the surface properties of EVs to improve the targeting efficiency. For instance, expressing αvβ3 integrins on EVs allows them to enhance the conduction ability of paclitaxel [62]. Additionally, Thapa et al. [63] summarized an innovative therapeutic strategy using EVs as drug delivery tools to integrate two different anticancer drug candidates, namely, tumor necrosis factor-related apoptosis-inducing ligand (TRAIL) and miR-335, to overcome resistance to sorafenib. Pharmaceutical companies have also developed commercialized autologous EV products. CK Exogene, a Korean biotech company which is currently developing EV-based anti-cancer drugs for liver cancer using the aforementioned drug candidates (i.e., TRAIL and miR-335), developed a patented technology (10-2020-0062365) for the mass production of EVs, overcoming the low yield challenge [64]. Another autologous bone marrow mesenchymal stem cell-derived EV product from Brianstorm, MSC-NTF (Nur Own®), has also been developed [65]. However, none of the EV-related products for clinical use have yet been approved by the U.S. Food and Drug Administration.

Drug loading into EVs can be achieved through two methods: pre-secretion and post-secretion. In the pre-secretion method, parental cells are co-incubated with the “drug” (usually with transfection reagents), and EVs are obtained from conditioned media in cultured cells or from biological tissues or fluids [66]. The drug is then sorted into EVs by active or passive means. Parent cells are modified by bioengineering to secrete large amounts of specific EVs. As one of the most readily available primary cells, mesenchymal stem cells (MSCs) are widely used as the parent cells for engineered EVs due to their high EV productivity, tropism, and homing effect [67]. As for ncRNAs, they are often chosen as candidates for the pre-secretion method [68]. However, the drug delivery efficiency of EVs in this method is relatively low [69]. To increase the delivery efficiency of EVs, specific components can be artificially loaded into the vesicles. For example, Li et al. [70] fused CD9, a characteristic protein of EVs, with an RNA-binding protein, HuR, to specifically enrich miR-155 in EVs.

Other approaches transfected the target ncRNA directly into the parental cells. Lou et al. [71] constructed miR-199a-modified adipose tissue-derived MSCs (AMSC-199a) by means of miR-199a lentivirus infection and obtained engineered EVs, which were found to be effective in transferring miR-199a to HCC cells. Engineered EVs expressing miR-199a-significantly increased the sensitivity of HCC cells to adriamycin *in vitro* and also significantly enhanced the antitumor effect of adriamycin on HCC *in vivo*. Similarly, miR-122-releasing plasmids were transfected into AMSC to generate miR-122-enriched EVs. These EVs induced G0/G1 phase arrest and apoptosis, increasing the sensitivity of HCC cells to chemotherapy [72]. Semaan et al. electroporated human brain endothelial cells with miR-214 to generate miR-214-enriched EVs. These EVs significantly increased the inhibition of anticancer agents (oxaliplatin or sorafenib) on HCC cells [73]. In addition to ncRNAs, which inherently have the ability to regulate gene expression, EVs were also being explored as carriers for delivering genome editing tools like CRISPR/Cas9. Yao et al. developed a novel aptamer-ABP system to actively enrich CRISPR RNPs into EVs for delivery [74]. Finally, siGRP78-modified bone marrow MSC EVs were reported to inhibit HCC growth and invasion both *in vivo* and *in vitro*, to sensitize sorafenib-resistant HCC cells, and to reverse their drug resistance [75].

On the other hand, post-secretion drug delivery involves isolating and purifying EVs before loading drugs into them. This method is more frequently used for large-scale industrial production. The loading of hydrophobic small molecules, such as curcumin [76] and paclitaxel [77], can be accomplished by simple incubation. Various methods (e.g., electroporation, extrusion, and sonication) have also been used to load hydrophilic molecules and larger molecules [78]. While this method is relatively simple and efficient in terms of loading drugs, additional purification steps are required in order to remove unencapsulated drugs, which may compromise the integrity of EVs.

Several successful examples of drug delivery using this method have been demonstrated in cancer therapy. For instance, Zhang et al. [20] loaded adriamycin or sorafenib into erythrocyte-derived EVs, which significantly inhibited the growth of hepatocellular carcinoma cells *in situ* in mice. Du et al. [79] loaded ferroptosis inducers (erastin, Er) and photosensitizer rose bengal (RB) into EVs (Er/Rb@Exos CD47) through an ultrasound technique, which demonstrated effective anti-tumor efficacy *in vitro* and *in vivo*. The transfection and enrichment of miR-335-5p into fibroblast-derived extracellular vesicles resulted in inhibition of HCC growth, proliferation, invasion, and apoptosis after intratumoral injection [80]. Other studies have shown that the delivery of EVs loaded with specific ncRNAs (e.g., miR-125b [81], miR-338-3p [82], miR-451a [83], miR-744 [84]) can significantly inhibit HCC growth by modulating a range of signaling pathways.

Despite these advances, the production of pharmaceutical EVs remains a major challenge for the industry. Further optimization of the process is needed for more efficient cargo loading, which would increase the drug dose within a single EV and reduce the number of EVs required. This would, in turn, improve therapeutic efficacy and facilitate clinical translation. To achieve this, more research is needed to reveal the mechanisms behind EV-mediated delivery and targeted uptake.

## 4. Therapeutic EVs for Immunomodulation in HCC

The tumor microenvironment (TME) plays a crucial role in cancer development and progression, and EVs have been found to mediate intercellular communication within the TME [85]. Dendritic cell-derived EVs (DEVs) are particularly promising as immunotherapeutic agents against cancer, as they contain functional MHC peptide complexes and other immunostimulatory components, and function as antigen-presenting entities. DEVs are suggested to promote an immune-cell-dependent tumor rejection response [86]. Studies have shown that DEVs can enhance the anti-tumor response by promoting the activation of CD8+ T cells and remodeling the TME [87]. For example, EVs released from antigen-presenting cells (e.g., DC cells or B cells) have been found to induce and maintain antitumor immune responses [88].

In addition, tumor-derived EVs also stimulate antitumor immune responses and act as antigen-presenting vesicles that deliver tumor-associated antigens (TAA) to dendritic cells. The TAA carried by the EV is readily taken up by dendritic cells, enabling efficient presentation of the antigens on MHC molecules to homologous T cells. Inspired by the concept of chimeric antigen receptor-T (CAR-T) cell therapy, researchers have stimulated dendritic cells with tumor antigens. MHC-antigen complexes can be presented by DEVs as CAR, which trigger T cell activation and effective anti-tumor immunity *in vitro* and *in vivo* [89]. By means of engineered anti-CD3 and anti-EGFR on DEVs to promote the binding of T cells to cancer cells, investigators have also shown that these EVs can be an effective and safe alternative to CAR-T cells, which are difficult to store and transport [90].

EVs engineered as tumor immunomodulators constitute an attractive approach for developing anti-tumor therapeutics due to their ability to act as tumor immunomodulators. One example is exoASO-STAT6, a precision drug candidate that uses EVs to selectively deliver antisense oligonucleotides to disrupt STAT6 signaling in tumor-associated macrophages (TAM) and induce antitumor immune responses [91]. Preclinical data released by Codiak BioSciences Inc. demonstrates the potent antitumor activity of its product candidate, exoASO-STAT6. Currently, a clinical trial using exoASO-STAT6 (CDK-004) in patients with advanced HCC or liver metastases is underway (NCT05375604). Another study found that EVs from the sera of HCC patients contained significantly less HMGN1 (high-mobility group nucleosome binding protein 1) than EVs from healthy individuals. Therefore, the researchers loaded a functional short peptide of HMGN1-N1ND onto the surfaces of EVs and successfully transported the N1ND molecule into dendritic cells to enhance their activation and immunogenicity [92]. Similarly, EVs exogenously loaded with immunomodulatory cyclic dinucleotides (CDN) were found to be 100–200 times more potent than free CDN in human peripheral blood mononuclear cells. Intravenous administration of CDN-EVs significantly improved immune surveillance and antitumor activity in preclinical models of HCC [93]. Another promising therapeutic strategy is to reprogram EVs to overexpress antibodies against PD-1, PD-L1, or CTLA-4. This reprogramming can improve the efficacy of immune checkpoint inhibitors (ICIs) in HCC patients [94]. In recent years, immunotherapy methods like atezolizumab (PD-L1 inhibitor) plus bevacizumab (VEGF inhibitor) have emerged as promising treatment options for HCC. Together, these have demonstrated efficacy in clinical trials and have been approved in some regions for the treatment of advanced HCC. However, specific links with EVs have not yet been illuminated by existing research.

Moreover, EVs can increase and modulate the immune response, making them a potential strategy for designing new vaccine formulations. EVs have the potential to activate granulocytes or NK cells and interact with CD8, CD4, and B cells to demonstrate antigen-specific immune responses [95]. Jesus et al. [96] suggested that EVs could enhance protective immune responses in vaccine development when coexisting with antigens. As they presented, EVs were isolated from a lipopolysaccharide endotoxin-stimulated human monocyte line (THP-1) as potential vaccine adjuvants and combined with hepatitis B recombinant antigen (HBsAg) solutions or suspensions composed of HBsAg-loaded poly-ε-caprolactone-chitosan nanoparticles. The combined EVs induced a humoral immune response similar to that of HBsAg. Figure 1 depicts a diagrammatic representation of engineered extracellular vesicles and their role in modulating the immune microenvironment in HCC.

In conclusion, both natural and engineered EVs show promise as immunotherapeutic agents against HCC and other types of cancer. However, further research is needed to develop safer and more effective anti-tumor therapies using these EVs.

Despite the significant progress in EV research, clinical translation still faces several limitations. *In vitro* EV-based diagnostics suffer from low sensitivity and high heterogeneity among different subpopulations. To validate EV cargoes as cancer biomarkers, faster and more convenient detection platforms are required.

*In vivo* intervention with EVs faces additional challenges, including immune exclusion, which is a primary concern in clinical translation. Although homologous and heterologous sources of EVs are unlikely to cause immune exclusion [97], many controversies still exist regarding the *in vivo* function and safety of heterologous EVs when used as delivery vehicles. For example, MSC-derived EVs may indirectly accelerate the progression of existing tumors due to their anti-inflammatory and immunosuppressive effects [98]. EVs from cells infected with viruses like HIV and HCV may transmit viral proteins and RNA to normal cells, promoting infection and chronic inflammation. There is also a risk of altering the orientation of membrane proteins when EVs are subject to drug loading and other membrane-disrupting manipulations, which may lead to immune attack and subsequent adverse reactions [99]. Similarly to blood products, the corresponding norms for the allogeneic transfer of EVs need to be emphasized. While autologous EV products can mitigate this risk, their production is limited to small-scale academic studies due to the high cost. Large-scale pharmaceutical production of autologous EVs is not yet feasible. The individualized nature of autologous EV products also makes their manufacture and quality control time-consuming, posing a challenge in terms of timely treatment for patients. Rational selection of parental cells, assessment of immunological and oncogenic effects, and continuous monitoring of the risk of viral contamination are necessary before EVs can be translated into safe and effective clinical therapeutic products [100].

Storage stability is another issue of concern. Storage at 4 °C can cause aggregation and damage to EV structures [101]. The recommended storage method for EVs is at −80 °C, but this method is expensive and may alter their biological activity [102]. Lyophilization is a more cost-effective method that allows EVs to be stored at room temperature without losing their key properties, such as vesicle size, protein content, structural integrity, biological activity, and pharmacokinetics of the transported molecules [103]. However, lyophilization requires the use of freeze-drying agents, which can be harmful to cells. GelMA hydrogels is another promising method for preserving EVs at room temperature [104]. Further research is required in order to assess the safety and efficacy of these methods in clinical settings.

Moreover, the isolation, purification, and qualification of EVs are critical procedures that should not be overlooked. Due to the innovative nature of EV therapies, there is a lack of internationally accepted standards for their clinical grading, production quality control, and application. The Minimal Information for Studies of Extracellular Vesicles 2018 (MISEV2018) [13] can improve the standardization of EV studies and the reproducibility of analysis results. However, the yield and quality of subpopulations of EVs by different isolation and purification methods can vary greatly. Therefore, it is necessary to form a specification and reach a consensus on the qualification of EVs and establish a standard operating procedure [98]. As technology continues to evolve, the analysis of vesicular biomolecules based on the single-particle level may become possible, which will aid in identifying and purifying biologically active subpopulations of EVs and determining the efficacy of EVs as therapeutic agents.

Finally, although progress has been made in modifying EVs, there is no guarantee that the engineered EVs will still target as they are expected to in the complex *in vivo* environment. Therefore, targeting modification of EVs remains a key task to be tackled.

## 5. Conclusions

The global market for EV products is anticipated to experience significant growth. However, the majority of EV-related studies are still in the laboratory research phase. To support the practical application of EVs in clinical diagnosis and disease treatment, it is crucial to conduct multi-center and large-sample clinical trials. Over the next 10 years, with the advancement of various technologies and the improvement of research standards, it is anticipated that there will be a significant acceleration in the translation of EVs from the laboratory to practical clinical applications. This golden era will be facilitated by basic research on the formation and mechanism of action of EVs, as well as continuous breakthroughs in the engineering modification, delivery, and mass production of EVs. Possible directions for further research include identifying subtle interactions between exosomes and the tumor microenvironment, investigating the possibility of cargo alterations in exosomes for targeted therapies, and investigating exosome-based immunotherapy approaches.

## Figures and Tables

**Figure 1 medicina-59-01599-f001:**
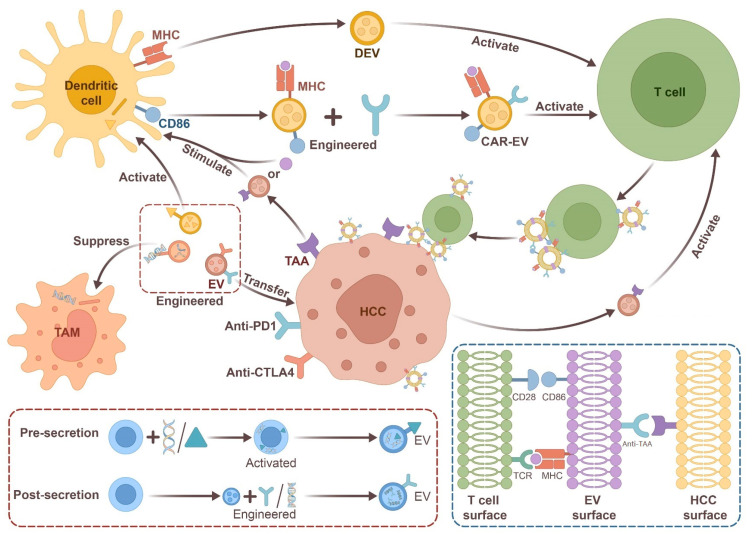
A schematic illustration of EVs engineered as tumor immunomodulators of HCC; HCC-derived EVs are known to incite immune cell activation by transporting related proteins (TAA), while engineered EVs also exert an effect on the immune microenvironment by conveying diverse signaling molecules. The cellular binding mediated by CAR-EV-induced T-cell activation upon recognition of HCC is highlighted in the dashed box on the lower right. In addition, the generative patterns of pre-secretory EVs and post-secretory EVs are demonstrated in the lower left dashed box. (HCC: hepatocellular carcinoma cell; DEV: dendritic cell-derived EVs; TAA: tumor-associated antigens; CAR-EV: chimeric antigen receptor-EV; TAM: tumor-associated macrophage; MHC: major histocompatibility complex). Translating EV Research to the Bedside: Challenges and Limitations.

**Table 1 medicina-59-01599-t001:** EVs as biomarkers for the diagnosis and prognosis of HCC.

Molecule Type	Identity of EV-Derived Molecules	Source of EVs	Effects	Diagnosis/Prognosis	Reference
miRNA	miR-21	serum/plasma	higher sensitivity in EVs than in serum alone; higher sensitivity and specificity than AFP in HCC diagnosis	Diagnosis	[23,25,27]
	miR-21-5p and miR-92a-3p	serum	better performance than AFP alone for HCC diagnosis	Diagnosis	[29]
	miR-122	serum	ideal for differentiating HCC from normal controls; can distinguish early-stage HCC from cirrhosis with significantly better accuracy than AFP alone	Diagnosis	[30,31]
	miR-21-5p and miR-144-3p	serum	have higher sensitivity and specificity than AFP in HCC diagnosis	Diagnosis	[28]
	miR-4661-5p	serum	good diagnostic performance at all stages of HCC, higher accuracy than serum AFP	Diagnosis	[32]
	miR-638	serum	significantly lower in the HCC group, and the degree of reduction is negatively correlated with cancer size and TNM stage	Diagnosis	[34]
	miR-331 and miR-23b	serum	miR-331 was significantly decreased while miR-23b was significantly increased in HCC patients after surgical resection	Diagnosis	[33]
	miR-221	serum	miR-221 expression was correlated with tumor size, cirrhosis, and tumor stage, and provided predictive significance for the prognosis of HCC patients.	Diagnosis/Prognosis	[42]
	miR-10b	serum	serve as diagnostic biomarkers for early HCC	Diagnosis	[36]
lncRNA	lncRNA-P11-583F2.2	plasma	higher sensitivity and specificity than AFP in diagnosing HCC	Diagnosis	[45]
	lncRNA JPX	plasma	diagnostic biomarker for HCC with high sensitivity and moderate specificity	Diagnosis	[46]
	ENSG00000258332.1 and LINC00635	plasma	combined with AFP, have a high AUC, sensitivity, and specificity	Diagnosis	[47]
	LncRNA ASMTL-AS1	plasma	LncRNA ASMTL-AS1 expression was associated with HCC stage, metastasis, and prognosis	Prognosis	[51]
	lncRNA-ATB	plasma	elevated lncRNA-ATB expression suggested shorter overall survival and progression-free survival	Prognosis	[48]
	LncRNA CRNDE	plasma	overexpressed in HCC patients with large, poorly differentiated tumors and advanced TNM staging, and was associated with poor prognosis	Prognosis	[50]
	LncRNA CTD2116N20.1 and lncRNA RP11538D16.2	Tissue/serum	shorter OS and PFS and contributed to poor prognosis	Prognosis	[49]
circRNA	hsa_circ_0004001, hsa_circ_0004123, and hsa_circ_0075792	plasma	higher diagnostic sensitivity and specificity: positively correlated with TNM stage and tumor size	Diagnosis	[54]
tsRNA	tRNA-ValTAC-3, tRNA-GlyTCC-5, tRNA-ValAAC-5, and tRNA-GluCTC5	plasma	differentially expressed in HCC patients and healthy donors	Diagnosis	[53]
piRNA	piR-1029; piR-15254; novel-piR-35395; novel-piR-32132; novel-piR-43597	serum	specific piRNAs are highly expressed in HCC patients and correlate with clinical features such as tumor size and lymph node metastasis	Diagnosis	[55]
protein	G3BP and PIGR	serum	elevated in the EVs of patients with HCC, and had a higher predictive efficacy for HCC than AFP	Diagnosis	[56]
	EpCAM, CD147, GPC3, and ASGPR1	tissue	evaluated for the detection of early HCC, and were able to differentiate early-stage HCC from cirrhosis	Diagnosis	[18]
	Smad3	serum	higher levels of SMAD3-EVs in HCC patients’ serum; positively correlated with disease stage and pathological severity and negatively correlated with disease-free survival	Diagnosis/Prognosis	[57]

## Data Availability

Not applicable.

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
