# Peer review of "Extracellular Vesicles in Hepatocellular Carcinoma: Progress and Challenges in the Translation from the Laboratory to Clinic"

_medicina, 2023, doi:10.3390/medicina59091599_

Round 1

Reviewer 1 Report

Authors tell about perspective models of using Extracellular Vesicles. Actually, it was and it is a mainstream in biomedicine research.  But this manuscript is included full information of this biological agent. 

I have some questions: 

1. If you tell about HCC, could you  please include information about standart treatment of this disease? And can we use both drugs of standart therapy, like atezolizumab plus bevacizumab, and Extracellular Vesicles? 

2. Could you please include some financial limitations of using in clinical practice of Extracellular Vesicles? 

3. What about toxic affect to the normal  cell?  can we protect them? 

Thank you for wonderful paper

Reviewer 2 Report

The authors have composed a comprehensive overview of the recent advancements in the realm of extracellular vesicles and their roles as both diagnostic biomarkers and therapeutic carriers in hepatocellular carcinoma, effectively addressing the associated opportunities and challenges. This manuscript sets itself apart from other contemporary review articles on the same subject due to its well-structured organization and the richness of detail and discussion. 

Minor Suggestions:

1. Could the authors and their affiliations be clarified?

2. I would recommend incorporating more examples and discourse on present studies that are seeking to enhance EV payload and pharmacokinetic curve. For instance, it would be beneficial to add discussion on optimizing the surface integrin and N-glycans.

3. The authors largely concentrate on EV-carried miRNA. It would be beneficial to add discussion on EV as genome editing carrier in the section titled "EVs: The Next Generation of Carriers for Drug Delivery."

Reviewer 3 Report

This study reviewed roles of EVs in HCC. Authors listed recent biomarkers including miRNAs, lncRNA, circRNA, tsRNA, piRNA and protein, in the diagnosis and prognosis of HCC. Besides, the potential of EVs as drug carriers was discussed as well in this study. In addition, authors reviewed intercellular communication which were mediated by EVs. It is informative to the research community. However, there are some concerns.

1.     Lack of complete review for HCC biomarkers, for example mir-665 was missing (https://doi.org/10.18632/oncotarget.20881). And mir-221 (https://doi.org/10.1016/j.bbrc.2011.01.111) et al.

2.     Since they are diagnostic biomarkers, diagnostic indices such as AUC, sensitivity and/or specificity would be included, that helps researchers better understand efficacies of these biomarkers. A meta-analysis might be better to show potential diagnostic roles if possible.

3.     A table listing approved and/or ongoing clinical trials with EVs as drug carriers could strengthen the part of “EVs, the next generation of carriers for drug delivery”.

It is OK.

Reviewer 4 Report

Dear Author,

The  review article titled “Extracellular Vesicles in Hepatocellular Carcinoma: Progress and Challenges in the Translation from the Laboratory to Clinic” is summarized well.  Please follow the instruction from the editor.

The review article titled “Extracellular Vesicles in Hepatocellular Carcinoma: Progress and Challenges in the Translation from the Laboratory to Clinic” by Rong Yan et al summarized well. It is well known that EV are small, membrane-bound structures that are released by various types of cells into the extracellular environment. These vesicles contain a variety of bioactive molecules, such as protein lips, nucleic acids (including microRNA), and even metabolites. EVs play critical roles in intercellular communication and are involved in various physiological and pathological process in the body. In this review article the author majorly focused on the recent studies in hepatocellular carcinoma. However, their roles in different biological processes are still being uncovered. Understanding the complexity of EV-mediated communication has the potential to open up new avenues for diagnostics, therapeutics, and insights into various diseases. Overall, this review article is considered for the publication with minor changes.

Minor changes:

1.       Author did not highlight recent studies on clinical trials in relationship with EV.

Minor editing of English language required

Reviewer 5 Report

Hepatocellular carcinoma (HCC), the most prevalent primary liver cancer, and exosomes are both covered by this review. Little vesicles known as exosomes are released by cells and are essential for cell-to-cell communication. Exosomes from cancer cells are thought to facilitate the development, invasion, and metastasis of cancer cells in the setting of HCC. These exosomes might also be used to diagnose and treat HCC; they serve as a flexible platform for both medication administration and diagnostics. In addition to reviewing exosomes' role in HCC proliferation, migration, and metastasis, the research looks at how they might be used for both diagnosis and therapy. The abstract also discusses the potential use of exosomes in HCC and notes the difficulties in this area of study.

The following additional information may be added to the text in order to enhance it:

1. Explain the clinical relevance of exosomes in HCC, focusing on how knowledge of their function in cell proliferation, migration, and metastasis might improve prognosis, therapeutic approaches, and patient outcomes.

2: Cargo Contents of Exosomes: Give further information about the precise molecular cargo transported by exosomes and how that contributes to their function in the development of HCC. This might include substances discovered in diverse investigations, including proteins, miRNAs, lncRNAs, and others.

3. Exosome Therapeutic Targeting: Discuss current research initiatives and prospective tactics targeted at exosome therapeutic targeting in HCC. Exosomes may be used as drug delivery systems, or strategies to prevent exosome-mediated drug resistance may be developed.

Mention the possibility of separating exosomes from different bodily fluids, such as blood, urine, and ascites, which may be important for non-invasive diagnostic and prognostic methods.

4. Discuss the benefits and drawbacks of employing exosome-based liquid biopsies as opposed to conventional tissue biopsies for the diagnosis and monitoring of HCC.

5. Highlight any current or finished clinical studies examining the use of exosomes as therapeutic agents in the treatment of HCC. Exosome-Based Therapeutics in Clinical Studies

6.Future Perspectives List possible directions for further study, such as figuring out the nuanced interactions between exosomes and the tumor microenvironment, examining the possibility of cargo alterations in exosomes for targeted treatment, and examining exosome-based immunotherapy methods.

7. Limitations: Recognize any gaps in the knowledge of exosomes in HCC, such as difficulties in their extraction, characterization, and cargo analysis, as well as the possibility of off-target consequences in therapeutic applications.

Extensive editing of English is needed

Round 2

Reviewer 3 Report

My concerns were addressed. Thank you.

Author Response

You're welcome! We're glad that your concerns have been addressed. Attached is the final version.

Reviewer 5 Report

The Authors have done all required corrections.

moderate corrections

Author Response

We're glad that your concerns have been addressed. Attached is the final version.
